# COMMON FEATURE LEARNING FOR ZERO-SHOT IMAGE RECOGNITION

## ABSTRACT

The key issue of zero-shot image recognition (ZIR) is how to infer the relationship between visual space and semantic space from seen classes, and then effectively transfer the relationship to unseen classes. Recently, most methods have focused on how to use images and class semantic vectors or class names to learn the relationship between visual space and semantic space. The relationship established by these two methods is class-level and coarse-grained. The differences between images of the same class are ignored, which leads to insufficiently tight relationships and affects the accurate recognition of unseen classes.To tackle such problem, we propose Common Feature learning for Zero-shot Image Recognition (CF-ZIR) method to learn fine-grained visual semantic relationships at the image-level. Based on the inter class association information provided by class semantic vectors, guide the extraction of common visual features between classes to obtain image semantic vectors. Experiments on three widely used benchmark datasets show the effectiveness of the proposed approach.

## 1 INTRODUCTION

In recent years, the development of general artificial intelligence has been rapid, and as a key link, zero-shot learning has received widespread attention. The key problem of zero-shot learning is how to infer potential knowledge between visual space and semantic space from seen categories, and then effectively transfer knowledge to unseen categories, finding corresponding semantic categories for the visual features of unseen class images, and achieving accurate class prediction of unseen class images.

Among the existing two types of ZSL methods, the generative model based method Wu et al. (2020); Chen et al. (2021a) learns the mapping from semantic space to visual space to generate visual features of unseen categories, thereby transforming the ZSL task into a traditional image classification task. This article believes that this method does not fundamentally solve the zero-shot problem and requires a large amount of computational resources, which is inconsistent with the original intention of the zero-shot problem. Another ZSL method, based on embedding methods Radford et al. (2021); Chen et al. (2022b); Shen et al. (2022); Wang et al. (2022), typically learns a common representation space between visual space and semantic space, where visual features and semantic vectors are projected onto the common representation space, enabling knowledge transfer from seen categories to unseen categories. However, most of these embedding based methods rely on image visual features and class semantic vectors to establish visual semantic connections, ignoring the fine-grained inter class association information provided by attributes.

It is worth noting that most methods focus on how to use images and class semantic vectors or class names to learn the relationship between visual space and semantic space, and the relationships established by these two methods are class level and coarse-grained. The differences between images of the same class are ignored, which leads to insufficient closeness and affects the accurate recognition of unseen classes.

Considering the above issues, this paper proposes a Common Feature learning for Zero-shot Image Recognition (CF-ZIR) method, which guides the extraction of common visual features between categories through attributes, and simulates expert scoring to obtain the degree to which an image contains a certain attribute, thus forming an image semantic vector. Specifically, by constructing a visual attribute cross domain dictionary, guidance is provided for the extraction of visual common

features by attributes. At the same time, the semantic vectors of images obtained based on common visual features are constrained to be similar to the semantic vectors of their respective categories, ensuring the effectiveness of common visual features. Finally, a fine-grained visual semantic cross domain dictionary is constructed based on image visual features and image semantic vectors to better capture the fine-grained associations between class independent visual and semantic information, thus achieving high-precision zero-shot image classification tasks.

Our contributions in this paper are summarized as follows:

- We propose the Common Feature learning for Zero-shot Image Recognition (CF-ZIR) method, which breaks new ground by discerning fine-grained visual-semantic relationships at the image level. This method leverages inter-class association information from class semantic vectors to guide the extraction of common visual features, leading to more nuanced image semantic vectors.

- CF-ZIR introduces a dual-layer embedding method, two layers of embeddings were established between visual-attribute and visual-semantic, respectively.

- A large number of experiments have been conducted to demonstrate that the CF-ZIR proposed in this chapter has achieved significant performance improvements on three benchmark datasets.

The remainder of this paper is organized as follows. Section 2 introduces related work. Section 3 introduces the methodology of CF-ZIR. Section 4 gives experimental results on three typical ZSL benchmark datasets. The conclusion is given in Section 5.

## 2 RELATED WORK

Zero-shot learning (ZSL) emerged from the challenge introduced by Larochelle et al. Larochelle et al. (2008), which questioned how to recognize images with limited labeled or unlabeled data. Lampert et al. Lampert et al. (2009) further propelled interest in ZSL within the image recognition community by introducing the Animals with Attributes (AwA) dataset, built on the concept of utilizing unlabeled data.

ZSL diverges from conventional image recognition by enabling the identification of new class images not encountered during model training, thus offering potential for numerous practical applications.Existing ZSL techniques can be broadly categorized into generative and embedding-based approaches.

Generative methods typically address ZSL by generating samples of unseen classes to train classifiers. Various methods leveraging Generative Adversarial Networks (GANs) Goodfellow et al. (2014) and other generative models have been proposed Xian et al. (2018); Ji et al. (2019); Han et al. (2021); Wu et al. (2020); Zhao et al. (2022); Chen et al. (2021a); Radford et al. (2021). Xian et al. Xian et al. (2018) presented a conditional GAN-based Mirza & Osindero (2014) approach where the discriminator was trained with class attribute classification loss, and the generator employed class attributes to produce visual features. Ji et al. Ji et al. (2019) suggested a dictionary-based method to generate pseudo-images for unseen classes, learning a dictionary for each seen class and generating pseudo-images for unseen classes by combining seen class dictionaries based on attribute distances. Recent generative ZSL methods, such as SDGN Wu et al. (2020) and FREE Chen et al. (2021a), focus on enhancing the discriminative power of generated visual features using constraints like feature refinement and self-supervised learning. However, generative methods tend to be more complex and computationally intensive than embedding-based approaches.

Embedding-based methods Jiang et al. (2018); Chen et al. (2022b); Shen et al. (2022); Yang et al. (2022); Wang et al. (2022) generally map visual features and semantic attributes into an embedding space and use distance metrics to find the closest class attributes to unseen class images. Jiang et al. Jiang et al. (2018) introduced CDL, an embedding-based method, which creates structured embeddings and aligns visual and semantic spaces by training a coupled dictionary with visual prototypes and class attributes. Chen et al. Chen et al. (2022b) proposed a mutual semantic distillation network that builds visual-semantic embeddings from regional visual features and attribute features. Shen et al. Shen et al. (2022) introduced a spherical ZSL method that measures similarities in a spherical

embedding space. Wang et al. Wang et al. (2022) incorporated local image information through a fully pixel-to-attribute embedding approach.

However, existing embedding-based methods concentrate on leveraging images and class semantic vectors or class names to establish the correlation between the visual and semantic domains, including CLIP Radford et al. (2021) et.al cross-modal large model. These approaches typically construct relationships at a class-level, which are broad and do not account for fine-grained details. As a result, they overlook the variations among images within the same class. This oversight can lead to a lack of closeness in the learned relationships, thereby adversely impacting the precision of recognizing classes that were not seen during the training phase.

In this paper, we introduce the Common Feature learning for Zero-shot Image Recognition method, which delves into the fine-grained visual-semantic relationships at the image level. Utilizing inter-class association cues from class semantic vectors, CF-ZIR enhances the extraction of common visual features across classes, resulting in more discriminative image semantic vectors.

## 3 METHODOLOGY

Firstly, the task ZSL is described formally in Section 3.1. Then, the proposed framework is briefly introduced in Section 3.2. Finally, the details of training the hierarchical coupled dictionary and recognizing the unseen image are described in Section 3.3 and Section 3.4 respectively.

### 3.1 PROBLEM FORMULATION

The task ZSL can be described as: given a seen class sample-label set $\{(y_i^s, h_i^s)\}_{i=1}^{N^s}$, where $y_i^s$ is a sample of seen class, and $h_i^s \in \mathcal{H}^s$ is its corresponding class label, the goal of ZSL is to predict the corresponding class label $h_j^u \in \mathcal{H}^u$ for an unseen sample $y_j^u$. It should be emphasized that the label set of seen classes $\mathcal{H}^s$ and that of unseen classes $\mathcal{H}^u$ are disjoint, i.e., $\mathcal{H}^s \cap \mathcal{H}^u = \emptyset$. Each class (both seen and unseen) is provided with a class attribute vector as available auxiliary information.

### 3.2 OVERALL FRAMEWORK

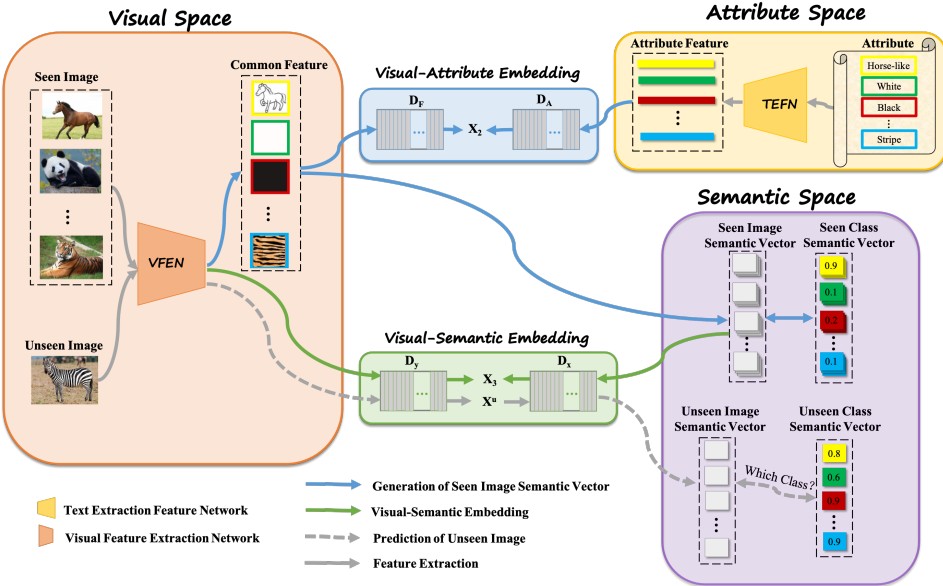

Figure 1: Framework of Common Feature Learning Zero-shot Image Recognition method. It shows the recognition performed in the semantic space based on the image-level coupled dictionary.

As shown in Fig. 1, the cross domain dictionary learning model for common feature perception proposed in this paper can be divided into two stages: visual attribute embedding stage and visual

semantic embedding stage. The gray solid line represents the feature extraction process, the blue solid line represents the model construction process in the visual attribute embedding stage, the green solid line represents the model construction process in the visual semantic embedding stage, and the gray dashed line represents the class prediction process for unseen class images.

In the visual attribute embedding stage, seen class images are first subjected to a visual feature extraction network (e.g. Res Net101) to extract visual features. The dictionary trained based on the visual features of seen class images can serve as common visual features between categories. By designing a visual attribute cross domain dictionary, the relationship between visual space and attribute space is established, which constrains the expression of common visual features and attribute features in the embedding space to be the same. In other words, common visual features are the corresponding expressions of attribute features in the visual space. Therefore, the sparsity coefficient obtained from the reconstruction of the common visual feature dictionary atom for a seen class image can describe the degree to which each attribute is included in the seen class image, that is, to obtain the semantic vector of the seen class image. By constraining the semantic vector of the seen class image to be similar to the semantic vector of the image's class, the accuracy of the common visual feature dictionary is ensured.

In the visual semantic embedding stage, the relationship between visual space and semantic space is established by constraining the seen class visual features and the seen class image semantic vectors to have the same expression in the embedding space. This relationship is class independent, so it can be generalized from the seen class to the unseen class, ensuring the model's recognition ability on the unseen class.

On the basis of the above two stages of model construction, there are three ways to predict the categories of unseen class images, namely: recognition in visual space, recognition in embedding space, and recognition in semantic space. Fig. 1 shows the process of semantic space recognition. unseen class images obtain visual features through feature extraction networks, generate semantic vectors of unseen class images through a visual semantic cross domain dictionary, and then find the closest class to the unseen class image by calculating the distance between the image semantic vector and each unseen class semantic vector.

### 3.3 TRAINING OF COMMON FEATURE LEARNING ZERO-SHOT IMAGE RECOGNITION METHOD

The proposed method trains the model through two stages, including the establishment of a visual-attribute coupled dictionary and the establishment of a visual-semantic coupled dictionary.

**Visual-Attribute Embedding**

At the first stage, based on a single dictionary learning model, a dictionary corresponding to the visual features of a seen class image is trained, and the semantic vector of the image is generated. The formula of the loss function is as follows:

$$\mathcal{L}_{ag}(\mathbf{F}, \mathbf{X}^s) = \|\mathbf{Y}_v - \mathbf{F}\mathbf{X}^s\|_F^2 + \lambda\|\mathbf{X}^s - \mathbf{Z}\|_F^2, \tag{1}$$

where $\mathbf{Y}_v \in \mathbb{R}^{M_v \times N^s}$ is the visual feature matrix of seen images, $M_v$ is the dimension of feature, and $N^s$ is the number of seen images, $\mathbf{F} \in \mathbb{R}^{M_v \times K}$ is the common visual feature dictionary, $\mathbf{F}$ contains $K$ dictionary atoms, each atoms is the description of each attributes in visual space, $\mathbf{X}^s \in \mathbb{R}^{K \times N_s}$ is the semantic feature matrix of seen images which describes the degree an image contains an attribute, $\mathbf{Z} \in \mathbb{R}^{K \times N_s}$ is the class semantic matrix of seen images.

The first constraint in Eq. 1 can reconstruct image visual features from the common visual feature dictionary $\mathbf{F}$ and the semantic feature matrix $\mathbf{X}^s$, while the second constraint the generates semantic feature matrix $\mathbf{X}^s$ that are close to the corresponding class semantic matrix $\mathbf{Z}$, $\lambda$ is a balance parameter used to adjust the contribution of the two constraints.

Extracting common visual features between categories based on attribute feature constraints, where each attribute feature corresponds to a common visual feature between categories. The formula of the loss function is as follows:

$$\mathcal{L}_{cf}(\mathbf{F}, \mathbf{D}_F, \mathbf{D}_A, \mathbf{X}_r) = \|\mathbf{F} - \mathbf{D}_F\mathbf{X}_r\|_F^2 + \mu\|\mathbf{A} - \mathbf{D}_A\mathbf{X}_r\|_F^2, \tag{2}$$

where $\mathbf{A} \in \mathbb{R}^{M_a \times K}$ is the attribute feature matrix extracted by the feature extraction network for attribute phrases, $\mathbf{D}_F \in \mathbb{R}^{M_v \times K}$ is the visual dictionary of the visual-attribute coupled dictionary,

$\mathbf{D}_A \in \mathbb{R}^{M_v \times K}$ is the attribute dictionary of the visual-attribute coupled dictionary, $\mathbf{X}_r \in \mathbb{R}^{K \times K}$ is the common description of attribute features and visual features in the embedding space, $\mu$ is a balance parameter used to adjust the contribution of the two constraints.

Totally, the loss function of the visual-attribute coupled dictionary learning stage is as follows:

$$\mathcal{L}_{att} = \mathcal{L}_{ag} + \alpha \mathcal{L}_{cf}, \tag{3}$$

where $\alpha$ is a balance parameter used to adjust the contribution of the two constraints.

The objective of optimization is to minimize the loss function $\mathcal{L}_{att}$. The variables to be solved include the common visual feature dictionary $\mathbf{F}$, the visual dictionary $\mathbf{D}_F$, the attribute dictionary $\mathbf{D}_A$, the semantic feature matrix $\mathbf{X}^s$, the common description matrix $\mathbf{X}_r$.

**Visual-Semantic Embedding**

At the second stage, align seen class images with seen class image semantic vectors by constructing a visual semantic cross domain dictionary pair. The corresponding formula is as follows:

$$\mathcal{L}_d(\mathbf{D}_y, \mathbf{D}_x, \mathbf{X}_e) = \|\mathbf{Y}_v - \mathbf{D}_y\mathbf{X}_e\|_F^2 + \eta\|\mathbf{X}^s - \mathbf{D}_x\mathbf{X}_e\|_F^2, \tag{4}$$

where $\mathbf{D}_y \in \mathbb{R}^{M_v \times L}$ is the visual dictionary of visual-semantic coupled dictionary, $\mathbf{D}_x \in \mathbb{R}^{K \times L}$ is the semantic dictionary of visual-semantic coupled dictionary, $L$ is the number of dictionary atoms, $\mathbf{X}_e \in \mathbb{R}^{L \times N^s}$ is the description of seen image in embedding space.

The discriminant loss is adopted to further constrain the discriminability of cross domain dictionaries, the corresponding formula is as follows:

$$\mathcal{L}_q(\mathbf{Q}, \mathbf{X}_e) = \|\mathbf{H} - \mathbf{Q}\mathbf{X}_e\|_F^2, \tag{5}$$

where $\mathbf{Q} \in \mathbb{R}^{C^s \times L}$ is the discriminator, $\mathbf{H} \in \mathbb{R}^{C^s \times N^s}$ is the label matrix of seen class image.

Totally, the loss function of the visual-semantic coupled dictionary learning stage is as follows:

$$\mathcal{L}_{vs} = \mathcal{L}_d + \beta \mathcal{L}_q, \tag{6}$$

where $\beta$ is a balance parameter used to adjust the contribution of the two constraints.

The objective of optimization is to minimize the loss function $\mathcal{L}_{us}$. The variables to be solved include the visual dictionary $\mathbf{D}_y$, the semantic dictionary $\mathbf{D}_x$, the discriminator $\mathbf{Q}$, the description of seen image in embedding space $\mathbf{X}_e \in \mathbb{R}^{L \times N^s}$.

Details of the training process of CF-ZIR are shown in Algorithm 1. The Line 1 to Line 7 are the visual-attribute embedding process, which includes common feature extraction and image semantic vector generation. The Line 8 to Line 13 are the visual semantic embedding process, which involves learning a visual semantic cross domain dictionary to obtain a dictionary pair. The initialization of the dictionary and classifier in algorithm is based on the KSVD algorithm, while the initialization of other variables is in the form of a random matrix.

### 3.4 RECOGNITION OF UNSEEN IMAGE

Based on the proposed framework, ZSL task is performed by mapping the data in visual space and semantic space into a definite space using the hierarchical coupled dictionaries. The definite space can be chosen from visual space, embedding space and semantic space, i.e., recognition in the visual space, recognition in the embedding space and recognition in the semantic space. In the following formulas, we take the recognition using image-level coupled dictionary as an example, and the class-level ones are similar.

**Recognition in the Visual Space**

In order to perform recognition in the visual space, the unseen class attributes $\mathbf{P}_s^u$ is firstly mapped into the embedding space using the image-level semantic dictionary $\mathbf{D}_s^{image}$. The corresponding formula is as follows:

$$\arg\min_{\mathbf{X}^u} \|\mathbf{P}_s^u - \mathbf{D}_s^{image}\mathbf{X}^u\|_F^2, \tag{7}$$

where $\mathbf{X}^u \in \mathbb{R}^{L \times C^u}$ is the embedding-description of unseen classes.

---

**Algorithm 1:** Training of CF-ZIR.

---

**Input** : Seen class sample pairs $(\mathbf{Y}_v, \mathbf{H})$; Class attributes $\mathbf{A}$; Semantic matrix of seen image $\mathbf{Z}$; Hyperparameters $\lambda, \alpha, \beta, \mu$ and $\eta$; Number of visual-semantic coupled dictionary atoms $L$;

**Output** : Visual-Semantic Coupled dictionaries $(\mathbf{D}_y, \mathbf{D}_x)$;

*// Visual-Attribute Coupled Dictionary Learning*

1 Initialize $\mathbf{F}, \mathbf{D}_F, \mathbf{D}_A$;

2 **repeat**

3     Update $\mathbf{X}_s$ via minimizing Eq. (1);

4     Update $\mathbf{X}_r$ via minimizing Eq. (2);

5     Update $\mathbf{D}_F$ and $\mathbf{D}_A$ via minimizing Eq. (2);

6     Update $\mathbf{F}$ via Eq. (3);

7 **until** *maximum iteration*;

*// Visual-Semantic Coupled Dictionary Learning*

8 Initialize $\mathbf{D}_y, \mathbf{D}_x, \mathbf{Q}$;

9 **repeat**

10     Update $\mathbf{X}_r$ via minimizing Eq. (6);

11     Update $\mathbf{D}_y$ and $\mathbf{D}_x$ via minimizing Eq. (4);

12     Update $\mathbf{Q}$ via minimizing Eq. (5);

13 **until** *maximum iteration*;

---

Then, the visual-description of unseen class is computed using the image-level visual dictionary $\mathbf{D}_v^{image}$, i.e., $\mathbf{P}_v^{u\prime} = \mathbf{D}_v^{image}\mathbf{X}^u$.

Finally, the cosine distance is adopted to measure the distances between the unseen image $\mathbf{y}_v$ and the visual-description of unseen classes $\mathbf{P}_v^{u\prime}$, searching the class nearest to the unseen image. The corresponding formula is as follows:

$$\arg \min_{c \in \{1, \cdots, C^u\}} (\mathcal{M}(\mathbf{P}_v^{u\prime}[c], \mathbf{y}_v)), \tag{8}$$

where $\mathbf{P}_v^{u\prime}[c] \in \mathbb{R}^{M_v \times 1}$ indicates the visual-description of the $c$th unseen class, $\mathcal{M}(\cdot, \cdot)$ indicates the cosine distance between two vectors.

**Recognition in the Embedding Space**

To perform recognition task in the embedding space, both the unseen image $\mathbf{y}_v$ and the unseen class attributes $\mathbf{P}_s^u$ are mapped into the embedding space using the image-level coupled dictionary.

The embedding-description of unseen class is computed using Eq. 7. The embedding-description of unseen image is computed using the image-level visual dictionary $\mathbf{D}_v^{image}$. The corresponding formula is as follows:

$$\arg \min_{\mathbf{x}^u} \|\mathbf{y}_v - \mathbf{D}_v^{image}\mathbf{x}^u\|_F^2. \tag{9}$$

Then, the cosine distances between the two embedding-descriptions are measured, and the class nearest to the unseen image is searched in the embedding space. The corresponding formula is as follows:

$$\arg \min_{c \in \{1, \cdots, C^u\}} (\mathcal{M}(\mathbf{X}^u[c], \mathbf{x}^u)), \tag{10}$$

where $\mathbf{X}^u[c] \in \mathbb{R}^{L \times 1}$ indicates the embedding-description of the $c$th unseen class.

**Recognition in the Semantic Space**

To perform recognition task in the semantic space, the unseen image is firstly mapped into the embedding space using the image-level visual dictionary, shown as Eq. 9. Then, the semantic-description of the unseen image is computed by $\mathbf{y}_s = \mathbf{D}_s^{image}\mathbf{x}^u$.

The distance between each column of unseen class attributes $\mathbf{P}_s$ and semantic-description of the unseen image $\mathbf{y}_s$ are measured by computing the cosine distance, and the class nearest to the unseen image is searched. The corresponding formula is as follows:

$$\arg \min_{c \in \{1, \cdots, C^u\}} (\mathcal{M}(\mathbf{P}_s^u[c], \mathbf{y}_s)), \tag{11}$$

where $\mathbf{P}_s^u[c] \in \mathbb{R}^{M_s \times 1}$ indicates the attribute of the $c$th unseen class.

Table 1: Statistics for attribute datasets: aPY, AwA1, AwA2 in terms of the number of seen image (Image.S), the number of unseen image (Image.U), the dimension of class attribute (Attr.), the number of seen class (Seen) and the number of unseen class (Unseen).

| Dataset | Image.S | Image.U | Attr. | Seen | Unseen |
|---|---|---|---|---|---|
| aPY Farhadi et al. (2009) | 5,932 | 7,924 | 64 | 20 | 12 |
| AwA1 Lampert et al. (2009) | 19,832 | 5,685 | 85 | 40 | 10 |
| AWA2 Xian et al. (2019) | 23,527 | 7,913 | 85 | 40 | 10 |

## 4 EXPERIMENTAL RESULTS

We give experimental results in this section. We show the results on four benchmarks (Section 4.2). Then, we demonstrate the effectiveness and necessity of each part of the proposed model, including the unseen adaptation, image attribute generation, and image-level coupled dictionary (Section 4.3). Finally, we analyze the quality of generated image attributes (Section 4.4).

### 4.1 DATASETS

We perform experiments on four ZSL datasets including aPascal & aYahoo (aPY) Farhadi et al. (2009), Animals with Attributes 1 (AwA1) Lampert et al. (2009) and Animals with Attributes 2 (AwA2) Xian et al. (2019) to verify the effectiveness of the proposed method. The statistics of all datasets are shown in Table 1. To make fair comparisons, we use the class attribute, image feature, data splits provided by Xian et al. (2017). The image features are extracted by the 101-layered ResNet He et al. (2016). Value of hyperparameters $\lambda, \alpha, \beta, \gamma, \mu, \eta$ are selected in the set $\{0.001, 0.01, 0.1, 1, 10\}$. The average per-class top-1 accuracy is used to measure the performance of models.

The three widely used benchmarks are briefly introduced as follows:

- aPY contains 32 categories, including bird, cow, chair, bus, etc.. They belong to three major classes, i.e., animal, object and vehicle. Images and attributes in this dataset are collected from Yahoo and Pascal VOC.
- AwA1 is an animal dataset, which contains 40 seen classes, including antelope, beaver, tiger, elephant, etc., and 10 unseen classes, including sheep, seal, rat, bobcat, etc..
- AwA2 contains the same fifty animal categories as AwA1 dataset. Different from AWA1 dataset, AwA2 dataset provides images collected from public sources, all licensed for free use and redistribution.

### 4.2 COMPARISON WITH STATE-OF-THE-ART

As the results shown in Table 2, our proposed method belongs to embedding-based methods, and achieves competitive results compared with the more complicated generative methods. Among the embedding-based methods, CF-ZIR achieves the best accuracy on AwA2 dataset, and second accuracy on aPY and AwA1 datasets.

### 4.3 ABLATION STUDIES

In order to demonstrate the effectiveness of each component in CF-ZIR, including the discrimination loss and the learning of image-semantic coupled dictionary, we design several ablation experiments on the three datasets. The results of ablation experiments are shown in Table 3.

**Discrimination Loss:** By comparing the CF-ZIR w/o DL and CF-ZIR in Table 3 row 1 and row 3, adding the discrimination loss brings improvements on the three datasets, especially on the AwA2 dataset. This phenomenon indicates that using discriminative loss can constrain the discriminability of cross domain dictionary pairs, thereby facilitating the recognition of unseen class images.

**Visual-Semantic Alignment:** The row 2 and row 3 in Table 3 show the results of ablation experiments on whether visual-semantic alignment, the CF-ZIR w/o VSA and CF-ZIR. Note that instead

Table 2: Recognition accuracies (unit: %) of CF-ZIR vs comparative methods on aPY, AwA1 and AwA2 datasets. The best and second-best results are marked in **Red** and **Blue**, respectively. "*" indicates the methods of adopting attribute features.

| Type | Methods | aPY | AwA1 | AwA2 |
|---|---|---|---|---|
| Generative | CCSS Liu et al. (2018a) | 35.5 | 56.3 | 63.7 |
| | RAS-cGAN Zhang et al. (2019) | 40.1 | 67.4 | - |
| | LisGAN Li et al. (2019) | 43.1 | 70.6 | - |
| | EDE Zhang et al. (2020) | 20.4 | 70.1 | 66.5 |
| | ACGN Liu et al. (2021) | 44.4 | 69.2 | 69.7 |
| | CE-GZSL Han et al. (2021) | - | 71.0 | 70.4 |
| Embedding-based | GAZSL Zhang et al. (2018) | 41.2 | 68.3 | 70.2 |
| | DCN Liu et al. (2018b) | 43.6 | 65.2 | - |
| | CDL Jiang et al. (2018) | 43.0 | 69.9 | 68.2 |
| | HSVA Chen et al. (2021b) | - | 70.6 | - |
| | TransZero* Chen et al. (2022a) | - | - | 70.1 |
| | MSDN* Chen et al. (2022b) | - | - | 70.1 |
| | ERPCNet Li et al. (2022) | 43.5 | - | **71.8** |
| | HCDDL Li et al. (2023) | **50.6** | **71.8** | 70.8 |
| | IAAC-net Chen & Zhou (2024) | - | - | 70.7 |
| | RSR Liu et al. (2024) | 45.4 | - | 68.4 |
| | CF-ZIR | **48.0** | **71.5** | **72.0** |

Table 3: Results of ablation experiments, Discrimination Loss (DL), Visual-Semantic Alignment(VSA).

| Model | aPY | AwA1 | AwA2 |
|---|---|---|---|
| CF-ZIR w/o DL | 47.7 | 70.9 | 70.4 |
| CF-ZIR w/o VSA | 46.0 | 67.5 | 66.0 |
| CF-ZIR | **48.0** | **71.5** | **72.0** |

of using visual-semantic alignment, the semantic vectors of the unseen class images are generated based on the common visual feature dictionary learned in the first stage, and the class semantic vectors closest to these vectors are found to predict the categories of the unseen class images. We can see that using visual-semantic alignment outperforms without it on the three datasets, especially by 4.0% and 6.0% on AwA1 and AwA2 datasets, respectively.

## 4.4 FURTHER ANALYSIS

This subsection analyzes the quality of image semantic vectors generated by CF-ZIR. High quality image semantic vectors help to learn better visual semantic cross domain dictionary pairs, ensuring the accuracy of visual semantic mapping. Therefore, in order to analyze the quality of image semantic vectors generated by CF-ZIR more intuitively, we visualizes the seen visual features of images and their corresponding image semantic vectors separately. Using the unsupervised dimensionality reduction method t-SNE, project high-dimensional vectors into a two-dimensional space.

As shown in Fig. 2, compared with the visual feature distribution of the image in the top line, the semantic vector distribution of the image in the bottom line exhibits more obvious intra class clustering and inter class dispersion. This not only indicates that the image semantic vectors generated by CF-ZIR are reasonable and reliable, but also shows that the image semantic vectors are of high quality, which is conducive to the establishing of visual semantic alignment.

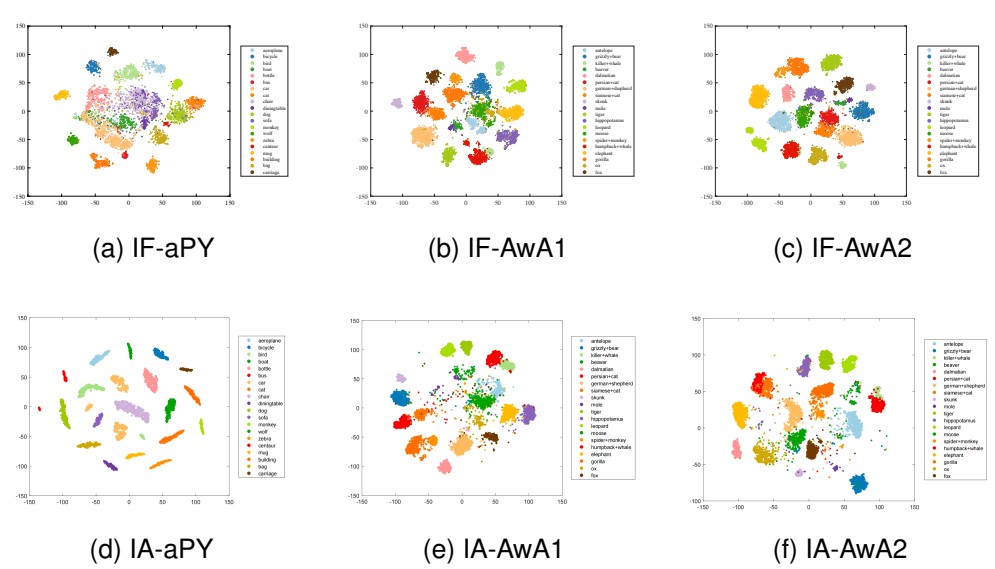

(a) IF-aPY       (b) IF-AwA1       (c) IF-AwA2

(d) IA-aPY       (e) IA-AwA1       (f) IA-AwA2

Figure 2: T-SNE visualization of the seen image features and the generated image attributes from aPY, AwA1 and AwA2 datasets (randomly selected several classes).

## 5   CONCLUSION

This paper proposed a common feature learning method for zero-shot image recognition (CF-ZIR), innovatively segmented into visual attribute and visual semantic embedding stages. By extracting common visual features and achieving cross-domain alignment between visual and semantic spaces, CF-ZIR adeptly captures the subtleties essential for recognizing classes not encountered during training. The method's efficacy is underscored by its exceptional performance on three major benchmark datasets, with ablation studies confirming the pivotal role of the discriminative term loss and cross-domain alignment in bolstering recognition accuracy.

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
