# OpenReview forum: "Common Feature Learning for Zero-shot Image Recognition"
_ICLR.cc/2025/Conference — Submitted to ICLR 2025_

### Official Review · Reviewer_3kEb · 2024-10-18

**Soundness:** 1
**Presentation:** 1
**Contribution:** 1
**Rating:** 3
**Confidence:** 4

**Summary:**

This paper proposes a novel method called Common Feature learning for Zero-shot Image Recognition (CF-ZIR) to address the challenge of learning fine-grade relationships between images and classes in Zero-Shot Learning. CF-ZIR leverages the inter-class association information from class semantic vectors to guide the extraction of common visual features between classes. The author conducted experiments on three datasets to attempt to validate the effectiveness of the proposed method.

**Strengths:**

It seems that the proposed approach is easy to follow.

**Weaknesses:**

This paper has major shortcomings in writing, method innovation, and experiment, so it is not able to be accepted, specifically:

This paper is hard to read. The meanings of many concepts are difficult to understand. For example, what is ‘the cross domain dictionary learning model’ mentioned in Section 3.2 and what are the particular meanings of ‘common feature’, ‘single dictionary learning model’ and ‘class semantic vectors’? The structure of the model and the calculation process are also hard to understand, what are ‘x2’ and ‘x3’ in Figure 1?

This paper is filled with grammatical and formatting errors, to the point that it's difficult to list them all. For example, in the abstract, 'zero-shot image recognition (ZIR)' should be 'Zero-shot Image Recognition (ZIR)', 'unseen classes.To tackle' should be ''unseen classes. To tackle', and ' Based on the inter class association information provided by class semantic vectors, guide the extraction of common visual features between classes to obtain image semantic vectors.' should be 'The inter-class association information provided by class semantic vectors guides the extraction of common visual features between classes to obtain image semantic vectors.'

Aligning images with attributes is a common idea in existing approaches (Modeling Inter and Intra-Class Relations in the Triplet Loss for Zero-Shot Learning (ICCV19) and Concept Bottleneck Models (ICML20)).

The author conducted experiments on only three small datasets, two of which are nearly identical (AWA1 and AWA2), and did not compare with multimodal pre-trained models such as CLIP.

**Questions:**

Please see Weaknesses.

---

### Official Review · Reviewer_Vq1T · 2024-10-29

**Soundness:** 2
**Presentation:** 2
**Contribution:** 2
**Rating:** 3
**Confidence:** 5

**Summary:**

This paper proposes common feature learning for zero-shot image recognition (CF-ZIR) method to learn fine-grained visual semantic relationships. This method leverages inter-class association information from class semantic vectors to guide the extraction of common visual features. Experiments on three datasets show the effectiveness.

**Strengths:**

+ Introducing dual-layer embedding method for zero-shot learning is reasonable.
+ The experimental results show the effectiveness of the proposed approach on conventional ZSL.

**Weaknesses:**

- The paper leverages inter-class association information from class semantic vectors to guide the extraction of common visual features. In fact, this issue has been defined as visual-semantic domain shift in previous work and has been discussed in several papers, such as [1], [2] and [3]. This paper does not discuss the differences with them.

- This paper only conducts experiments under the conventional zero-shot learning setting, lacking experiments in generalized zero-shot learning.

- Line 16, "The relationship established by these two methods is class-level and coarse-grained", what do these two methods refer to?

- Introduction and Related work sections do not discuss the latest research.

- The ablation experiments are insufficient to validate the claims of this paper, especially regarding zero-shot recognition in three spaces discussed herein.

- The paper discerns fine-grained visual-semantic relationships at the image level, but why are experiments not conducted on fine-grained datasets, such as CUB?

- There is a lack of hyperparameter analysis. The current hyperparameters are not convincing. Are they the same across all datasets?

- How does this method demonstrate advantages for attributes like 'small', 'hunter', and 'fast' on AWA2?

**Questions:**

See Weaknesses.

---

### Official Review · Reviewer_nakS · 2024-11-03

**Soundness:** 3
**Presentation:** 2
**Contribution:** 2
**Rating:** 5
**Confidence:** 5

**Summary:**

This paper mainly introduces a shared feature learning method (CF-ZIR) for zero-sample image recognition, which extracts shared visual features through attribute guidance, and uses category semantic vectors to guide the generation of image semantic vectors, so as to form the semantic representation of images. In addition, a dual-layer embedding method is proposed to establish fine-grained associations between visual-attributes and visual-semantics. Experimental results show that the CF-ZIR method achieves competitive performance on multiple datasets.

**Strengths:**

1. The dual-layer embedding mechanism is introduced to improve the recognition performance through fine-grained visual-semantic relationship, which exhibits a degree of innovation.
2. The paper is articulated with clarity and precision.

**Weaknesses:**

1. Although the experimental results are presented in the paper, the description of experimental settings, hyperparameter selection, training details and other aspects is not detailed enough, which may affect other researchers to reproduce the results.
2. The proposed method does not achieve the best results on multiple datasets.
3. Several references to the three datasets were made with inconsistent fonts.

**Questions:**

1. A large number of hyperparameters are given in this paper, but the selection of hyperparameters and the comparison of ablation experiments are not described much.
2. The improvement effect of the proposed method is not obvious, and the best results are not achieved on both aPY and AWA1 datasets. Please explain.
3. Please give the results of this model on the commonly used dataset CUB in ZSL, and compare with the recent methods.
4. In the absence of ablation experiments, multiple loss functions are given in the article. Please provide relevant ablation experiments so that I can understand their impact on the model results.

---

### Official Review · Reviewer_snW4 · 2024-11-06

**Soundness:** 1
**Presentation:** 1
**Contribution:** 1
**Rating:** 1
**Confidence:** 5

**Summary:**

This paper proposed to learn a common features for ZSL, which include two kind dictionary for the final prediction.

**Strengths:**

1. The proposed method is innovative in using a dual dictionary approach to improve class prediction accuracy.

2. The paper is well-structured, making it relatively easy to follow the methodology and experimental setup.

**Weaknesses:**

1. The motivation in the third paragraph of the introduction, stating that "most methods focus on how to use images and class semantic vectors or class names to learn the relationship between visual space and semantic space," is somewhat inaccurate. In fact, most methods utilize class-level attributes as semantic information. Some also incorporate embeddings of individual attributes to enhance fine-grained associations.

2. Section 3.2 is overly lengthy and has significant overlap with subsequent sections. The overview could be more concise to avoid redundancy and improve clarity.

3. There is confusion around the definitions of semantic space and attribute space. What is the intended difference? In ZSL, attributes are generally considered a type of semantic information, so a clearer distinction would improve understanding.

4. The paper lacks comparisons with recent methods and is limited in dataset diversity. Additional comparisons with recent approaches and experiments on more datasets, such as CUB, FLO, and SUN, would strengthen the evaluation.

5. More extensive experimentation is needed. The current paper includes only two tables and one figure, which is insufficient to support its claims. Particularly, experiments should cover both conventional ZSL and generalized ZSL.

This paper appears incomplete and requires substantial revision before submission. More work is needed to clarify motivations, streamline the methodology, and provide comprehensive experimental validation.

**Questions:**

Please see the `Weaknesses’ above.

---

### Meta-Review · Area_Chair_cB2g · 2024-12-23

**Metareview:**

The paper introduces the Common Feature learning for Zero-shot Image Recognition (CF-ZIR) method, which uses a novel dual dictionary approach to enhance class prediction accuracy by leveraging inter-class association information from semantic vectors. However, the paper suffers from several weaknesses, including a lack of motivation and context and inadequate experimental results and analysis (e.g., ablation study and hyperparameters). Other minors, such as redundancy in writing and numerous grammatical and formatting errors, could be addressed in future submissions.

**Additional Comments On Reviewer Discussion:**

The authors did not provide responses, and there is no discussion.

---

### Decision · Program_Chairs · 2025-01-22

Reject